# Knee Injury Detection using MRI with Efficiently-Layered Network (ELNet)

**Chen-Han Tsai**                                          MAXWELLTSAI@YAHOO.COM
*School of Electrical Engineering, Tel Aviv University, Israel*

**Nahum Kiryati**                                              NK@ENG.TAU.AC.IL
*The Manuel and Raquel Klachky Chair of Image Processing, School of Electrical Engineering, Tel-Aviv University, Israel*

**Eli Konen**                                    ELI.KONEN@SHEBA.HEALTH.GOV.IL
**Iris Eshed**                                   IRIS.ESHED@SHEBA.HEALTH.GOV.IL
**Arnaldo Mayer**                            ARNALDO.MAYER@SHEBA.HEALTH.GOV.IL
*Diagnostic Imaging, Sheba Medical Center, affiliated to the Sackler School of Medicine, Tel-Aviv University, Israel*

## Abstract

Magnetic Resonance Imaging (MRI) is a widely-accepted imaging technique for knee injury analysis. Its advantage of capturing knee structure in three dimensions makes it the ideal tool for radiologists to locate potential tears in the knee. In order to better confront the ever growing workload of musculoskeletal (MSK) radiologists, automated tools for patients' triage are becoming a real need, reducing delays in the reading of pathological cases. In this work, we present the Efficiently-Layered Network (ELNet), a convolutional neural network (CNN) architecture optimized for the task of initial knee MRI diagnosis for triage. Unlike past approaches, we train ELNet from scratch instead of using a transfer-learning approach. The proposed method is validated quantitatively and qualitatively, and compares favorably against state-of-the-art MRNet while using a single imaging stack (axial or coronal) as input. Additionally, we demonstrate our model's capability to locate tears in the knee despite the absence of localization information during training. Lastly, the proposed model is extremely lightweight ($< 1$MB) and therefore easy to train and deploy in real clinical settings.

**Keywords:** Knee Diagnosis, MRI, Deep Learning, ACL Tear, Meniscus Tear, Knee Injury, Medical Triage

## 1. Introduction

Magnetic Resonance Imaging (MRI) has long been considered the most robust knee examination tool available (Saeed, 2018). Its widespread use is partly due to its capability to capture detailed structures in the knee joint while remaining a non-invasive procedure (Crues et al., 1987; Boeree et al., 1991). Given its profound capabilities to capture the knee in three dimensions, MRI has become the tool-of-choice for radiologists in an extensive range of examinations such as knee osteoarthritis and internal derangement of the knee. (Hayashi et al., 2014; Arumugam et al., 2015). Considering the ever growing workload of musculoskeletal (MSK) radiologists, automated tools for patients' triage are needed, leading

to shorter delays in the reading of pathological cases. Several techniques have been proposed for this purpose. Štajduhar et al. (2017) presented a semi-automated approach that used support vector machines (SVM) to diagnose anterior cruciate ligament (ACL) injuries in the knee. In their work, an ROI is first manually extracted before being fed into the SVM for prediction. Liu et al. (2018) introduced a fully-automated cartilage lesion detection system by employing a CNN for segmentation followed by another CNN for patch classification. Although their network is trained end-to-end, the amount of manual labeling required to create the patch training set makes it an overwhelmingly cumbersome task. Bien et al. (2018) proposed an architecture that consists of three individual MRNets whose output are combined using logistic regression. An MRNet extracts a distinctive feature vector for each slice of the scan, stacks the vectors into a 2D array, max-pools the array to obtain a single vector, and performs classification by a fully connected layer with softmax activation. The backbone of the feature extractor is a pre-trained AlexNet (Krizhevsky et al., 2012).

In this work, we present an Efficiently-Layered Network (ELNet) architecture optimized for knee diagnosis using MRI. The main contribution of this work is a novel slice feature extracting network that incorporates multi-slice normalization along with BlurPool down-sampling. The proposed methods will be detailed in Section 2, followed by quantitative and qualitative experimental results in Section 3. Conclusion and future work will be given in Section 4.

## 2. Methods

The ELNet architecture is illustrated in Figure 1 and the details are listed in Table 1. The backbone of ELNet's design centers around *Block* modules. Inspired by ResNet (He et al., 2016), we define a *Block* as a sequence of:

$$[\text{2D Convolution} \rightarrow \text{Multi-slice Normalization} \rightarrow \text{ReLU activation}]$$

*Blocks* are designed to allow for non-linearities in the network, and they may be repeated while ensuring equal input and output dimensions. A skip connection is added between the input and output, allowing better optimization of the network. The first two *Blocks* are repeated twice with $4K$ and $8K$ channels, and the remaining *Blocks* are fixed with $16K$ channels.

Each *Block* is followed by another 2D Convolution and ReLU activation, and they serve to increase channel dimension. The spatial height and width are reduced using a BlurPool layer. Eventually, in the final layer of the feature extractor, 2D max-pooling is applied to obtain a $16K$-dimensional feature vector for each MRI slice. Max-pooling is consecutively applied to obtain a single $16K$-dimension feature vector that combines feature information across slices. Dropout is performed before feeding into a fully-connected layer with two output logits, and the final probability $p(y|x)$ is computed by softmax (Goodfellow et al., 2016).

In the following two subsections, we detail two innovative features of ELNet: the use of multi-slice normalization, and BlurPool.

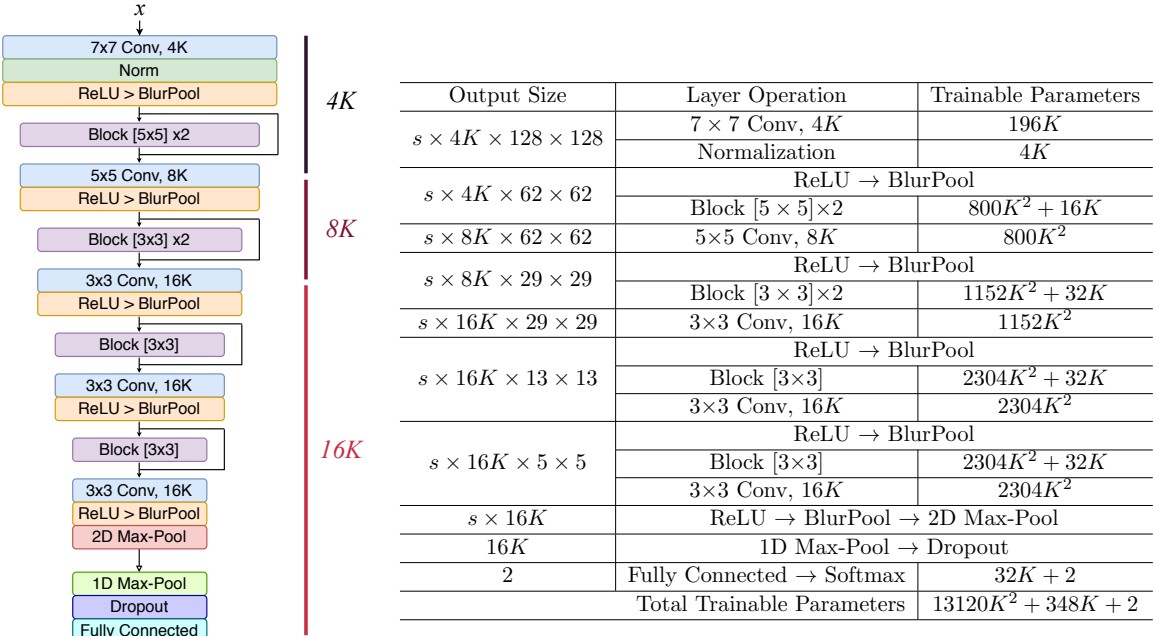

Figure 1: ELNet Design

Table 1: ELNet architecture in detail

| Output Size | Layer Operation | Trainable Parameters |
|---|---|---|
| $s \times 4K \times 128 \times 128$ | $7 \times 7$ Conv, $4K$ | $196K$ |
| | Normalization | $4K$ |
| | ReLU $\to$ BlurPool | |
| $s \times 4K \times 62 \times 62$ | Block $[5 \times 5] \times 2$ | $800K^2 + 16K$ |
| $s \times 8K \times 62 \times 62$ | $5 \times 5$ Conv, $8K$ | $800K^2$ |
| | ReLU $\to$ BlurPool | |
| $s \times 8K \times 29 \times 29$ | Block $[3 \times 3] \times 2$ | $1152K^2 + 32K$ |
| $s \times 16K \times 29 \times 29$ | $3 \times 3$ Conv, $16K$ | $1152K^2$ |
| | ReLU $\to$ BlurPool | |
| $s \times 16K \times 13 \times 13$ | Block $[3 \times 3]$ | $2304K^2 + 32K$ |
| | $3 \times 3$ Conv, $16K$ | $2304K^2$ |
| | ReLU $\to$ BlurPool | |
| $s \times 16K \times 5 \times 5$ | Block $[3 \times 3]$ | $2304K^2 + 32K$ |
| | $3 \times 3$ Conv, $16K$ | $2304K^2$ |
| $s \times 16K$ | ReLU $\to$ BlurPool $\to$ 2D Max-Pool | |
| $16K$ | 1D Max-Pool $\to$ Dropout | |
| $2$ | Fully Connected $\to$ Softmax | $32K + 2$ |
| | Total Trainable Parameters | $13120K^2 + 348K + 2$ |

## MULTI-SLICE NORMALIZATION

We propose two possible variants of multi-slice normalization: a first one based on *layer normalization* (Ba et al., 2016), and a second one based on *contrast normalization* (Ulyanov et al., 2016). Let's assume a feature representation $x^{(i)} \in \mathbb{R}^{S \times C \times H \times W}$ from some layer $i$ in the network (usually a 2D-convolution), where $S$ is the number of slices in the MRI sequence, $C$ is the number of channels in the representation, and $H, W$ are the spatial height and width of the representation. The network applies a normalization on $x$ (omitting $i$ for simplicity) by computing the appropriate mean and variance.

In the *layer normalization* variant, the mean $\mu_s$ and variance $\sigma_s^2$ are computed from $x$ for each slice $s$ ($1 \le s \le S$). In *contrast normalization*, the mean $\mu_{sc}$ and variance $\sigma_{sc}^2$ are computed for each slice $s$ and also for each channel $c$ ($1 \le c \le C$) (Figure 2 a-c). Using the computed mean and variance, $x$ is standardized into $\hat{x}$. An affine transform is applied to $\hat{x}$ to obtain the normalized output $y$. The normalization process is expressed by Equation (1) for *layer normalization* and Equation (2) for *contrast normalization* respectively:

$$\hat{x}_n = \frac{x_n - \mu_n}{\sqrt{\sigma_n^2 + \epsilon}} \to y_n = \gamma \hat{x}_n + \beta \qquad \forall n : 1 \to N \tag{1}$$

$$\hat{x}_{nc} = \frac{x_{nc} - \mu_{nc}}{\sqrt{\sigma_{nc}^2 + \epsilon}} \to y_{nc} = \gamma \hat{x}_{nc} + \beta \qquad \forall n : 1 \to N, c : 1 \to C \tag{2}$$

Parameters $\gamma$, $\beta$ ($C$ dimensional vectors) are learned independently for each normalization layer. Typically, $\gamma, \beta, \epsilon$ are initialized to **1**, **0**, and 1e-8 respectively.

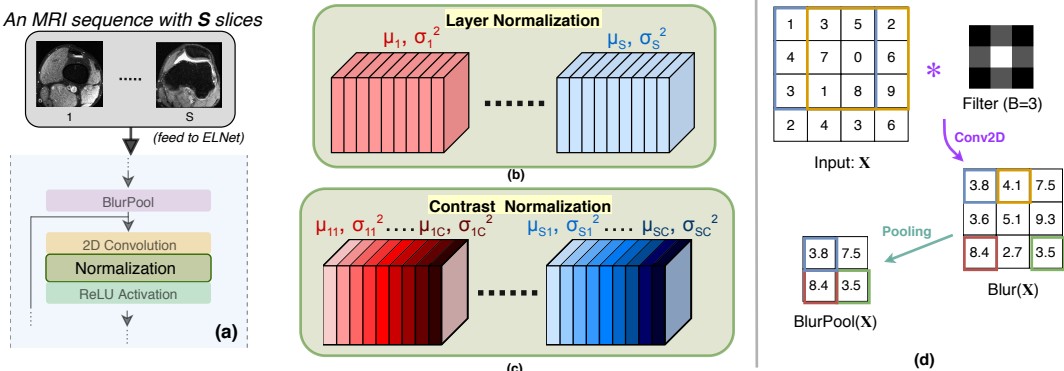

Figure 2: **(a)** An MRI sequence fed as input to ELNet, and an illustration of an ELNet Block. **(b&c)** Our proposed multi-slice normalization: Layer normalization and Contrast normalization (multi-slice norm standardizes slice-wise unlike batch norm which standardizes channel-wise) **(d)** BlurPool example: Input $X$ is convolved with binomial filter (kernel $B = 3$) to obtain an anti-aliased representation Blur($X$). Pooling is then applied to obtain BlurPool($X$).

## BLURPOOL

In the work of Zhang (2019), a BlurPool operation was proposed to mitigate the shift-variance phenomenon observed in modern CNN architectures where max-pooling is often utilized. BlurPool functions by first applying an anti-aliasing filter (binomial filter with kernel size $B$ and stride 1) to the input representation, then strided pooling is applied to obtain the pooled feature map (see Figure 2d). The resulting representation is therefore a pooled version of the blurred input representation, and a more detailed analysis is available in the paper (Zhang, 2019).

### 2.1. Training Pipeline

As suggested by Nyúl and Udupa (1999), we perform histogram-based intensity standardization according to the training set statistics, thus enabling similar-valued pixels to be associated with the relevant tissue type. In addition, we perform randomized data augmentations to each series which includes translation, horizontal flip, scaling, and minor rotations up to $\pm 10$ degrees around the center of the volume. For volumes captured in the axial and coronal orientations, we apply an additional random rotation of a multiple of 90 degrees to the volume. Finally, all the images are resized to $256 \times 256$ before entering the network.

Aside from data augmentation, we implement oversampling to compensate for dataset imbalance. For each pathology, we select the minority class samples (allowing repeats) from our training set and apply augmentations on them until the number of minority class samples (along with their augmented copies) equals the number of majority samples.

We train ELNet using standard cross-entropy loss (Goodfellow et al., 2016). Optimization can be done using a simple grid-search over relevant hyperparameters such as learning rate, choice of multi-layer normalization, BlurPool kernel sizes, dropout rate, etc.

## 3. Experiments

### 3.1. Datasets

**MRNet Dataset.** The MRNet Dataset contains 1,370 knee MRI examinations that were carried out at the Stanford University Medical Center. Each case was labeled according to the presence/absence of an anterior cruciate ligament (ACL) tear, a meniscus tear, or other signs of abnormalities in the corresponding knee. Each exam was randomly assigned either to the training, validation, or test set (Bien et al., 2018). It should be noted that each exam may contain multiple labels (e.g. an exam labeled positive for abnormality and ACL tear indicates other forms of abnormality in addition to an ACL tear).

The provided dataset includes, for each case, corresponding axial, coronal and sagittal MRI acquisitions. As reported by Bien et al., a sagittal T2-weighted series, a coronal T1 weighted series, and an axial proton density weighted series were selected for this dataset. Each image is of size $256 \times 256$ and the number of slices ranges between 17- 61 (mean 31 and standard deviation 7.97). The MRNet Dataset is currently the largest public labeled knee MRI dataset.

**KneeMRI.** The KneeMRI dataset collected at the Clinical Hospital Centre Rijeka, Croatia by Štajduhar et al consists of 917 exams labeled with ACL conditions in the corresponding knee. For each exam, the ligament condition was classified as either healthy (690 exams, 75.2%), partially injured (172 exams, 18.8%), or completely ruptured (55 exams, 6%). Each assessment corresponds to a T1-weighted sagittal MRI series, containing $320 \times 320$ or $290 \times 300$ images. The number of images in each series ranges between 21-45 (mean 31 and standard deviation 2.27). The dataset was divided into 10 strata with similar distributions, and we perform stratified sampling for evaluation.

### 3.2. Training

**MRNet Dataset.** In the MRNet dataset, we were provided with three imaging orientations per examination. For the three pathologies, we trained three separate ELNet's with $K = 4$, and the network weights were initialized uniformly by choosing the best random seed between 0-4 (He et al., 2015). Based on experiments, we selected coronal images for detecting meniscus tears, and axial images for detecting ACL tears and abnormalities. Contrast normalization yielded the best results for detecting meniscus tears, and layer normalization for detecting ACL tears and abnormalities (notice the correspondence between the selected multi-slice normalization and image modality.) Each model was trained using Adam with a learning rate between 1e-5 and 3e-5 for 200 epochs, taking roughly 1.5 hours (Kingma and Ba, 2014).

**KneeMRI Dataset** With the KneeMRI dataset, we perform 5-fold cross validation using eight out of the ten strata, and validation using the remaining two. Similar to the MRNet Datset, we train an ELNet with K=2 using SGD+Momentum for 200 epochs and the training time is roughly an hour for each fold (Sutskever et al., 2013).

| Architecture | Pathology | Accuracy | Sensitivity | Specificity | ROC-AUC | MCC |
|---|---|---|---|---|---|---|
| MRNet | Meniscus Tear | 0.735 | 0.827 | 0.662 | 0.826 | 0.489 |
| | ACL Tear | 0.9 | 0.907 | 0.894 | 0.956 | 0.769 |
| | Abnormality | 0.883 | 0.947 | 0.64 | 0.936 | 0.628 |
| ELNet | Meniscus Tear | 0.88 | 0.86 | 0.89 | **0.904** | **0.745** |
| | ACL Tear | 0.904 | 0.923 | 0.891 | **0.960** | **0.815** |
| | Abnormality | 0.917 | 0.968 | 0.72 | **0.941** | **0.736** |

Table 2: Evaluation Statistics between MRNet and ELNet on the MRNet validation set

By choosing K=2, and K=4 for the ELNet architectures, our trained model involves 53,178, and 211,314 trainable parameters respectively. In relation to AlexNet ($\sim$61M trainable parameters), ELNet (with K=4) contains $288\times$ fewer parameters than AlexNet. In comparison with MRNet, ELNet with K=4 contains $866\times$ less parameters, and ELNet with K=2 contains $1147\times$ less parameters. Each trained model was saved using standard PyTorch format. Model sizes are 850kB and 435kB for K=4 and K=2 respectively. Our experiments were perfomed on an NVIDIA GTX 1070 8GB GPU.

### 3.3. Evaluation

**MRNet Dataset.** We evaluate ELNet's performance using the validation set provided by the MRNet dataset (since the test set is not publicly available), and we compare it with the MRNet model proposed and trained by Bien et al. Although they evaluated their models primarily using the ROC-AUC, we perform a more thorough analysis by considering additional metrics that are just as significant, such as Sensitivity and the Matthew Correlation Coefficient (MCC). The evaluation results are presented in Table 2 and the ROC is plotted in Figure 3 (a-c), where we can observe noticeably higher MCC of the ELNet model.

**KneeMRI Dataset.** We evaluate ELNet using a 5-fold cross-validation scheme in detecting injuries in the ACL. The evaluation metrics following the 5-folds are shown on figure 3 (d-g); we highlight the lowest value in for each metric in red. In the original paper, Štajduhar et al trained an SVM and reported an AUC of 0.894 using 10-fold cross-validation. Bien et al reported an AUC of 0.911 on a particular train/valid/test set split using a pre-trained MRNet. In our experiment, we obtain an average AUC of 0.913 from the 5-folds, with three of the five folds exceeding 0.92 and the highest being 0.924. Moreover, we observe just minor variations in multiple performance metrics across folds; this demonstrates our model's robustness despite limited data and a highly unbalanced distribution.

### 3.4. Ablation Studies

This section aims to compare ELNet performance when multi-slice normalization and Blur-Pool are replaced with batch normalization (Ioffe and Szegedy, 2015) and max-pooling. The objectives are the three pathologies presented in the MRNet dataset, and the best results following the modified ELNet designs are listed in Table 3. Stemming from the fact that batch normalization induces an undesired standardization for each channel of feature

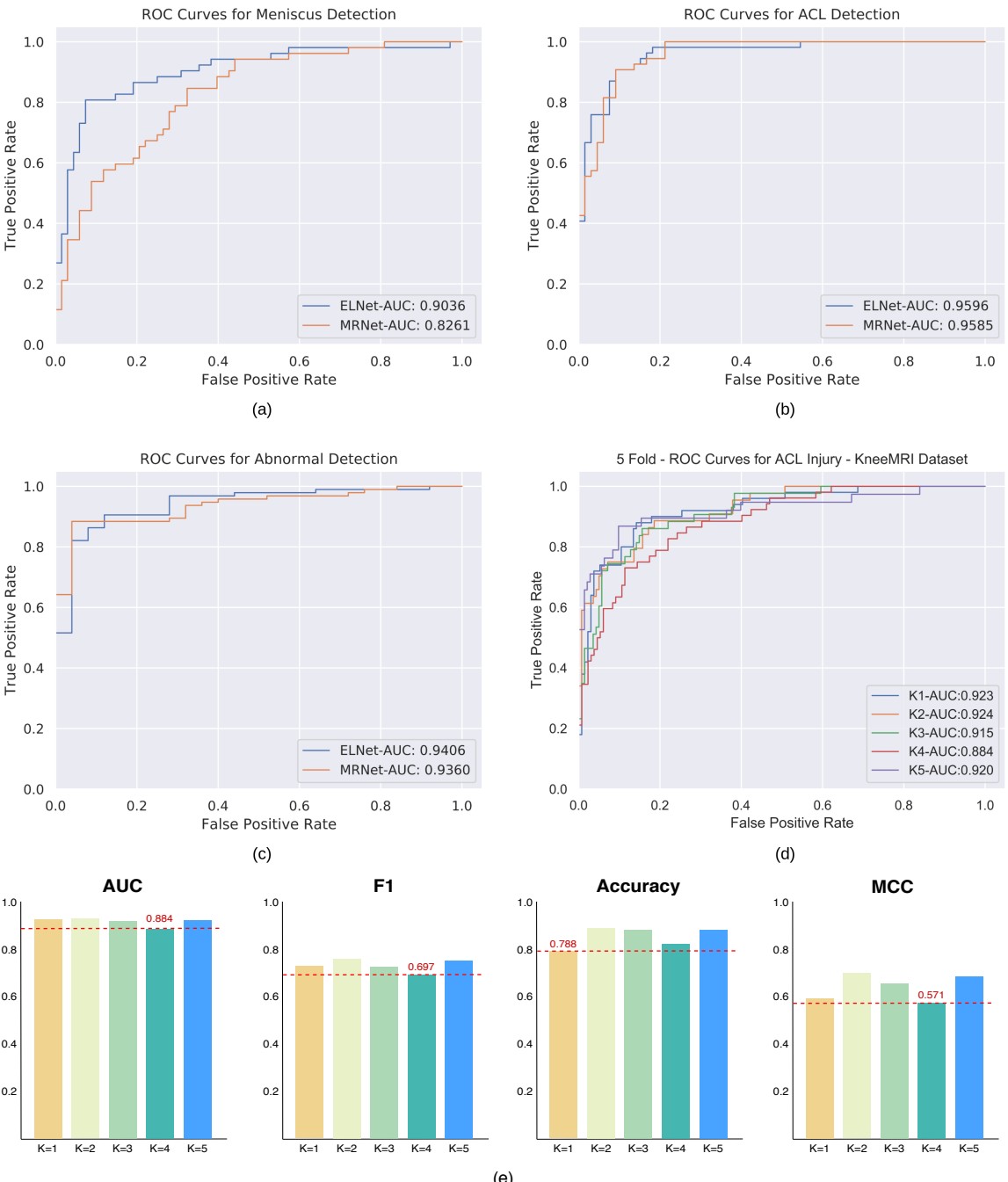

Figure 3: **MRNet Dataset:** (a-c) Comparision of ELNet and MRNet ROC     **KneeMRI Dataset:** (d) ELNet ROC's obtained from 5-fold cross-validation (e) ELNet metrics following 5-fold cross-validation

| ELNet Configuration | Meniscus Tear | | ACL Tear | | Abnormalities | |
|---|---|---|---|---|---|---|
| (K=4) | ROC-AUC | MCC | ROC-AUC | MCC | ROC-AUC | MCC |
| Multi-Slice Norm + BlurPool | **0.904** | **0.745** | **0.960** | **0.815** | 0.941 | **0.736** |
| Batch Norm + BlurPool | 0.751 | 0.391 | 0.871 | 0.530 | 0.841 | 0.440 |
| Multi-Slice Norm + MaxPool | 0.848 | 0.534 | 0.923 | 0.633 | **0.943** | 0.557 |
| Batch Norm + MaxPool | 0.7972 | 0.403 | 0.906 | 0.693 | 0.880 | 0.312 |

Table 3: Comparison of ELNet performance when multi-slice normalization and BlurPool are replaced with batch normalization and max-pool. The ROC-AUC and MCC of the best performing model (one for each pathology) of each ELNet configuration is reported.

representations across all slices, the feature extractor (designed to extract per-slice features) would no longer process each slice independently, and degraded performance deems reasonable. Following our experiments, it is evident that the use of batch normalization aggravates ELNet performance. In practice, we observe network divergence during training after 10-15 epochs. To our surprise, ELNet with batch norm and max-pool delivered slightly improved performance when compared with ELNet with batch norm and BlurPool, but when BlurPool is paired with the intended multi-slice normalization, we observe an overall improvement in performance compared to max-pooling.

### 3.5. Model Interpretation

To understand how ELNet identifies certain attributes for diagnosis, we compute the Full-Gradient representation of ELNet using the FullGrad algorithm (Srinivas and Fleuret, 2019). FullGrad generates a heat-map that corresponds to parts of the input that most influence the output prediction. Conceptually, the generated heat-map should be "hotter" in areas indicating an injury and "cold" elsewhere.

To verify that ELNet is indeed performing diagnosis based on features in the given acquisition, we randomly selected one of the five cross validation splits and evaluated the trained ELNet from that split. Samples from the validation set were randomly selected from both classes, resulting in 9 cases containing ACL tear and 7 cases without. A board-certified MSK radiologist with 17 years of experience was asked to identify the most informative slice (slice containing the most area for which a tear resides) in a given series and furthermore indicate the region in the (most informative) slice corresponding to an ACL injury. The identical task was performed on the trained ELNet, and of the 9 cases that contain ACL tear, the trained ELNet's prediction of the most informative slice and tear region coincided with the radiologist's evaluation in 8 of the cases. Of the 7 cases where the ACL is intact, our model's prediction matched the radiologist's assessment in all 7 cases. In Figure 4, we present a few examples of the generated heat-maps.

### 4. Conclusion

In this work, we present ELNet, a unique CNN architecture optimized for knee injury detection. The novel integration of multi-slice normalization and BlurPool operations allow ELNet models to remain lightweight ($\sim$ 0.2M parameters, requiring single imaging stack, trained from scratch) while performing favorably against MRNet models ($\sim$ 183M

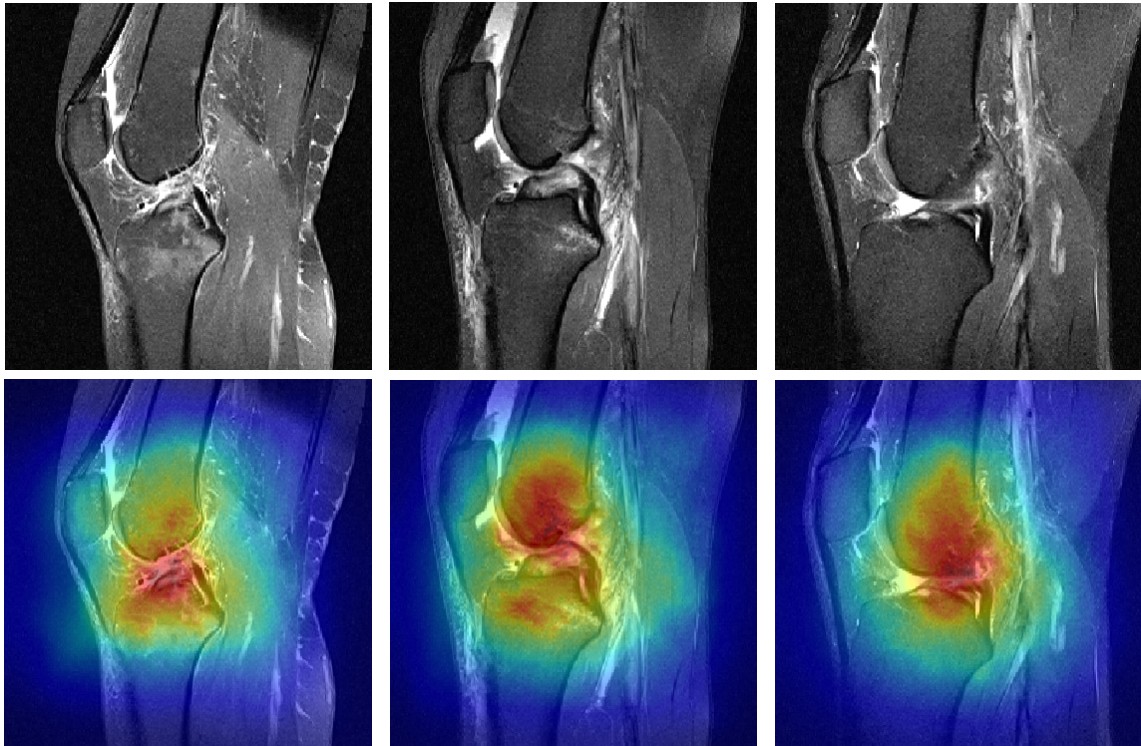

Figure 4: **Top:** Sample MRI slices containing ACL tears. **Bottom:** Full-Grad visualization computed using the above slices. "Hotter" areas indicate regions containing ACL tear.

parameters, requiring three imaging stacks, pretrained AlexNet) on the MRNet dataset. Cross-validation on the KneeMRI dataset have demonstrated consistent improved performance with ELNet models, proving the architecture to be robust regardless of a highly unbalanced distribution. In a clinical setting, where large number of cases await evaluation, our algorithm may be used for triage, improving workflow efficiency. In addition, by having our algorithm locate regions containing tears, radiologists can benefit by having the most significant slice presented first for each case.

Future work may include performance enhancement by incorporation of all three MRI volumes, axial, coronal and sagittal, if available. Further research is also needed to facilitate application of trained models on MRI data acquired using different scanners with various intensity scales. With the promising findings thus far, we believe ELNet may serve as a solid basis for future works involving knee injury triage.

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
