# OpenReview forum: "Knee Injury Detection using MRI with Efficiently-Layered Network (ELNet)"
_MIDL.io/2020/Conference — MIDL 2020_

### Official Review · AnonReviewer4 · 2020-03-13
**Method is well explained and the experiment for comparing with state-of-the-art is complete, but ablation study lacks.**

**Rating:** 2
**Confidence:** 3

**Summary:**

The paper proposed an interesting method (ELNet) to diagnose anterior cruciate ligament for knee MRI. By using multi-slice normalization and BlurPool. The cross-validation experiments for 2 different datasets show good improvement from previous state-of-the-art method. Hyper-parameter search is complete to get the best proposed model. But ablation study for multi-slice normalization and BlurPool lacks.

**Strengths:**

The paper is well written and describes an interesting and relatively novel approach to solving knee diseases.
The methods are well explained and results are well compared to previous state-of-the-art.
parameters are well searched to get the highest performance.
The key contribution is:
Different normalizaltion methods are used for different diseases to boost the performance.
BlurPool is used for the network.
The proposed network achieves higher AUC and MCC.


**Weaknesses:**

1. The purpose of BlurPool how it improves the model is  unclear. BlurPool is a pre-defined 3x3 kernel following by trided down-sampling, which may have been well used in the backbone. Ablation study lacks.
2. The proposed network applies different normalization for different diseases, but no results support the fact the for some disease, one kind of normalization is better than the other.
3. Lack of novelty. BlurPool is the only novelty of this paper, and layer-normalization/contrast-normaliztion acts more as a normalization search for different diseases.

**Justification Of Rating:**

The result shows good improvement for knee diseases in MRI. Two different datasets are evaluated. Hype-parameters are well searched, but more importantly, the paper lacks ablation study for the proposed two novelty. And It is not convincing to take into account multi-slice normalization as novelty.

**Paper Type:**

validation/application paper

**Questions To Address In The Rebuttal:**

1. The purpose of BlurPool. Maybe I missed, I didn't see the explanation about the purpose of BlurPool.
3. Lack of novelty. The is mainly because multi-slice normalization uses 2 existing normalization methods and it is trivial how different normalization method for different disease is chosen.

**Special Issue:**

no

---

> ### Author Response · Authors · 2020-03-24
> **Re: Method is well explained and the experiment for comparing with state-of-the-art is complete, but ablation study lacks.**
>
> We would like to thank the reviewer for an in-depth review of our work, and we appreciate the comments made.
>
> As suggested by the reviewer, we will add an ablation study to our final revision that investigates the effect our proposed multi-slice normalization compared with the other normalization methods (including Batch Norm) and the effects of Max-Pool compared with BlurPool performance (both jointly and independently) on the ELNet architecture.
>
> Regarding the questions addressed by the reviewer:
>
> > The purpose of BlurPool
>
> BlurPool is a pooling operation that mitigates the shift-variance phenomenon observed in CNN’s that employ Max-Pooling. There are various kernel sizes (2-7) for defining a BlurPool kernel and the (frozen) weights are that of a 2D Binomial Filter. From a signal processing perspective, pooling operations are equivalent as spatial sampling. By applying BlurPool, feature representations are first passed through a low-pass filter before being pooled, and this preserves shift-invariance for feature representations in the CNN. In the backbone of the MRNet feature extractor, only Max-Pooling was employed, and in the ELNet feature extractor, pooling operations were all performed using BlurPool.
>
> > Lack of novelty. The is mainly because multi-slice normalization uses 2 existing normalization methods and it is trivial how different normalization method for different disease is chosen.
>
> We agree with the reviewer that the multi-slice normalization utilizes 2 existing normalizations. However, contrast normalization is typically used for image stylization, and layer normalization is often used for NLP tasks. Given that 3D MRI images are the inputs to an ELNet, we believe that the unconventional use of contrast normalization and layer normalization (as opposed to the typical Batch Normalization)  for a classification task should be considered a novelty.
>
> For detecting meniscus tear, coronal images were selected, and contrast normalization was chosen. For detecting ACL tears and general abnormalities, axial images were selected and layer normalization was chosen. Although such choices were determined empirically, we believe that the reason certain multi-slice normalization works well for certain image stacks relates to the orientation the images were captured.
>
> In addition, our empirical results (will be added to the ablation studies) demonstrate that training diverges after 10-15 epochs whenever Batch Norm was applied. This relates to the fact that Batch Norm applies an undesired standardization for each channel of feature representations across all slices, and therefore, multi-slice normalization plays a critical role in optimizing ELNet models for their respective tasks.

---

### Official Review · AnonReviewer2 · 2020-03-13
**A Model for Diagnosing Knee Pathologies**

**Rating:** 3
**Confidence:** 3
**Recommendation:** Oral

**Summary:**

In this work, the authors purposed a new deep neural network architecture for detecting injuries/abnormalities in the knee. The main contribution of the work was adding a normalization step to the network, and learning the affine transformation parameters during the training. The normalization was followed by a BlurPool layer to solve the shift variance.

**Strengths:**

The paper is written very well, the implementation details are provided to help reproducing the results.
The method was tested on two different datasets, which is impressive. The results of the model was compared also to the state of the art.

**Weaknesses:**

From the following sentence, I understand that for each pathology, a different model was trained. If this is true, the model is not efficient.
“Contrast normalization yielded the best results for detecting meniscus tears, and layer normalization for detecting the remaining pathologies.”

**Detailed Comments:**

This sentence is not completely true: “a model with higher sensitivity is always preferred since better detection of true positives has always been the goal of automated diagnosis algorithms.”
A good model should have a high sensitivity as well as specificity.


**Justification Of Rating:**

The algorithm was explained very well. The results are also very nice. However, if different models were trained for predicting each parameter, not only training but also prediction would not be efficient.

**Paper Type:**

both

**Questions To Address In The Rebuttal:**

Please provide quantification for this sentence:
“In the majority of the cases provided, ELNet was able to indicate the most informative slice and has generated a heatmap that coincided with the region reported by the radiologist.”

What was the reason for choosing different values for K foe two datasets? Did these values obtained using cross validation?

**Special Issue:**

yes

---

> ### Author Response · Authors · 2020-03-24
> **Re: A Model for Diagnosing Knee Pathologies**
>
> We would like to thank the reviewer for a careful review of our work, and we appreciate the feedback provided.
>
> We agree with the reviewer’s remark regarding the statement we made “a model with higher sensitivity is always preferred since better detection of true positives has always been the goal of automated diagnosis algorithms”, and we will remove this statement from the final revision.
>
> The reviewer was correct in interpreting that an ELNet was trained for each pathology, and we will make sure to clarify this in our revision. When using the term “efficient”, we referred to our model as being memory efficient and computationally efficient when compared with the SOTA MRNet. Additionally, for each pathology in the MRNet dataset, our model was trained on a single imaging stack and compares favorably to MRNet which was trained on all three imaging stacks for each pathology.
>
> Regarding the questions addressed by the reviewer:
>
> > Please provide quantification for this sentence:
> “In the majority of the cases provided, ELNet was able to indicate the most informative slice and has generated a heatmap that coincided with the region reported by the radiologist.”
>
> Model evaluation by the radiologist was carried out by randomly selecting one of the five cross validation splits. Samples (confirmed by the radiologist) were randomly selected from both classes of the validation set (following the split) resulting in 9 cases containing ACL tear and 7 cases without. The model being evaluated was trained on the training set (following the split) and of the 9 cases that contain ACL tear, our model’s prediction of the most informative slice coincided with the radiologist's slice selection in 8 of the cases. Of the 7 cases where the ACL is intact, our model’s prediction matched the radiologist’s slice selection in all 7 cases. As noted by the reviewer, the details of the model evaluation will be added to our revision.
>
>
> > What was the reason for choosing different values for K for two datasets? Did these values obtained using cross validation?
>
> Yes, the values for K were determined empirically. Quantitatively, we selected a smaller K for ELNet’s trained on the KneeMRI dataset since the training set contains around 730 samples for each split. The training set for the MRNet dataset contains 1,130 samples, and so a slightly bigger model (bigger K) was selected to avoid underfitting.

---

### Official Review · AnonReviewer1 · 2020-03-13
**Knee Injury Detection using MRI with Efficiently-Layered Network (ELNet)**

**Rating:** 3
**Confidence:** 4
**Recommendation:** Poster

**Summary:**

The authors propose a lightweight CNN model (< 1 MB) for locating potential tears in the knee on MRI images. The main contributions are two normalization layers (layer and contrast normalization) for 3D sub-images and the application of BlurPool downsampling. Promising results are shown on two  knee datasets.

**Strengths:**

The paper is well written and easy to follow.

Even though the proposed model is lightweight (0.2M), it is shown to be on par or better than a recently published model called MRNet (183M parameters).

The selected application (discovering knee tear) seems to be clinically relevant.




**Weaknesses:**

It is not entirely clear how crucial the proposed multi-slice normalization and BlurPool layers are. An ablation study and comparison to established methods like batch normalization would have been valuable.

**Detailed Comments:**

The sub-figures in Figure 2 could be explained in more detail. What do the different colors and arrows mean for instance?

**Justification Of Rating:**

The method adopts approaches from the literature (instance normalization, BlurPool) and applies them to the problem of knee tear detection on 3D MRI data.

The paper is well written but would benefit from an ablation study to better understand the value of the individual layers in comparison to the standard approach using batch normalization.

The results and model size of the proposed approach are enticing.

**Paper Type:**

validation/application paper

**Questions To Address In The Rebuttal:**

I appreciate that an experienced board-certified MSK radiologist was asked to identify the most informative slice. How was "most informative" defined? What is the performance (quantitatively) of ELNet on finding this slice?

What is the intution/formal definition of MCC and why is it preferred compared to the ROC-AUC measure?

Why was the normalization performed in the slice direction only? Could it be of value in other spatial directions (e.g., in plane)?

Which statistical tests were applied to show statistical significance for the numbers in bold of Table 2?

Will the source code be made publicly available upon publication?


**Special Issue:**

no

---

> ### Author Response · Authors · 2020-03-24
> **Re: Knee Injury Detection using MRI with Efficiently-Layered Network (ELNet)**
>
> We would like to thank the reviewer for a detailed review of our work.
>
> As the reviewer suggested, we will add an ablation study to our final revision that investigates the effect of Batch Normalization compared to our proposed multi-slice normalizations and the effects of Max-Pool compared with BlurPool performance (both jointly and independently) on the ELNet architecture.
>
> The explanations for the sub-figures in Figure 2 will be updated with more details, and we thank the reviewer for pointing this out.
>
> Regarding the questions addressed by the reviewer:
>
> > How was "most informative" defined? What is the performance (quantitatively) of ELNet on finding this slice?
> The “most informative” slice refers to the image (in an MRI sequence) that contains the most area for which the tear resides.
>
> Model evaluation by the radiologist was carried out by randomly selecting one of the five cross validation splits. Samples (confirmed by the radiologist) were randomly selected from both classes of the validation set (following the split) resulting in 9 cases containing ACL tear and 7 cases without. The model being evaluated was trained on the training set (following the split) and of the 9 cases that contain ACL tear, our model’s prediction of the most informative slice coincided with the radiologist's slice selection in 8 of the cases. Of the 7 cases where the ACL is intact, our model’s prediction matched the radiologist’s slice selection in all 7 cases. As noted by the reviewer, the details of the model evaluation will be added to our revision.
>
> > What is the intuition/formal definition of MCC and why is it preferred compared to the ROC-AUC measure?
>
> The formal definition of the Matthew’s Correlation Coefficient is :
>
> $$ MCC = \frac{TP \times TN - FP \times FN}{\sqrt{(TP+FP)(TP+FN)(TN+FP)(TN+FN)}} $$
>
> Intuitively, it measures how “good” a classifier is at correctly predicting both the majority of the negative and positive cases. Given that the ROC-AUC refers to the area under the ROC curve (TPR and FPR are plotted for various classifier thresholds), the ROC-AUC is heavily influenced by the predicted positive class, making it an unreliable metric for evaluating class-imbalanced datasets (the case with the KneeMRI dataset). On the other hand, the MCC is invariant to class swapping, and is the preferred metric in the event of a class-imbalanced distribution for evaluating a classifier’s performance. An article by Chicco et al addresses the advantages of the MCC in more detail : https://doi.org/10.1186/s12864-019-6413-7
>
> > Why was the normalization performed in the slice direction only? Could it be of value in other spatial directions (e.g., in plane)?
> The reason normalization was only applied in the slice direction relates to the assumption that features were to be extracted independently for each slice prior to the 2D-Max-Pool. If normalization was performed in the plane direction (as in Batch Normalization), each channel $c (1 <= c <= C)$ of each slices’s feature representation would be standardized across all slices $s (1 <= s <= S)$. Doing so would imply dependence between slices when the designed feature extraction was intended to operate independently for each slice. Empirically speaking, models with Batch Normalization diverged after 10-15 epochs (will be added to the ablation study).
>
> > Which statistical tests were applied to show statistical significance for the numbers in bold of Table 2?
> The numbers bolded in Table 2 were intended to highlight the evaluation metrics that demonstrate a favorable comparison between ELNet and MRNet on the MRNet dataset, and we apologize for the confusion. To the reviewer’s inquiry, we performed a McNemar Test between the trained ELNet and MRNet on the three classification pathologies. We obtained a p-value of 0.009, 0.387, 0.99 for the classification of meniscus tear, general abnormalities, and ACL tear respectively. Thus, we may reject the null hypothesis that the two models' performances are equal for detecting meniscus tears, and we may not reject the null hypothesis in the case of detecting general abnormalities and ACL tears. As the reviewer mentioned, ELNet’s performance is on par with MRNet in detecting general abnormalities and ACL tear, and exceeds MRNet’s performance in detecting meniscus tear.
>
> > Will the source code be made publicly available upon publication?
> The source code will be made available pending authorization from our academic institution.

---

### Meta-Review · Area_Chair1 · 2020-04-07
**MetaReview of Paper117 by AreaChair1**

**Rating:** 3
**Recommendation For Accepted Papers:** Poster

**Metareview:**

The reviewers agree that this is a well-written piece of work. The authors achieve promising results while proposing a very lightweight model architecture with regard to memory consumption. There is some concern about the lack of ablation studies that could show the benefit of some of the design choices made by the authors. However, the use of some normalization layers typically used in different fields of deep learning and the application of blur pooling are showing interesting results in this application and warrant further discussion. The authors are encouraged to add the missing ablation experiments to their final paper version.

**Paper Type:**

validation/application paper

**Special Issue:**

no

---

### Decision · Program_Chairs · 2020-04-11

Accept